# Food Bioactive Compounds and Emerging Techniques for Their Extraction: Polyphenols as a Case Study

**DOI:** 10.3390/foods10010037

**Published:** 2020-12-24

**Authors:** José S. Câmara, Bianca R. Albuquerque, Joselin Aguiar, Rúbia C. G. Corrêa, João L. Gonçalves, Daniel Granato, Jorge A. M. Pereira, Lillian Barros, Isabel C. F. R. Ferreira

**Affiliations:** 1CQM—Centro de Química da Madeira, Campus Universitário da Penteada, Universidade da Madeira, 9020-105 Funchal, Portugal; joselin.aguiar@staff.uma.pt (J.A.); jluis@staff.uma.pt (J.L.G.); jorge.pereira@staff.uma.pt (J.A.M.P.); 2Departamento de Química, Faculdade de Ciências Exatas e da Engenharia da Universidade da Madeira, Campus Universitário da Penteada, 9020-105 Funchal, Portugal; 3Centro de Investigação de Montanha (CIMO), Instituto Politécnico de Bragança, Campus de Santa Apolónia, 5300-253 Bragança, Portugal; bianca.albuquerque@ipb.pt (B.R.A.); rubiacorrea@ipb.pt (R.C.G.C.); iferreira@ipb.pt (I.C.F.R.F.); 4REQUIMTE—Science Chemical Department, Faculty of Pharmacy, University of Porto, Rua Jorge Viterbo Ferreira N° 228, 4050-313 Porto, Portugal; 5Program of Master in Clean Technologies, Cesumar Institute of Science Technology and Innovation (ICETI), Cesumar University—UniCesumar, Parana 87050-390, Brazil; 6Food Processing and Quality, Natural Resources Institute Finland (Luke), Tietotie 2, FI-02150 Espoo, Finland; daniel.granato@luke.fi

**Keywords:** food bioactive compounds (FBCs), polyphenols, disease protection, cancer, cardiovascular diseases (CVDs), neurodegenerative diseases (NDs), bioavailability and bioaccessibility

## Abstract

Experimental studies have provided convincing evidence that food bioactive compounds (FBCs) have a positive biological impact on human health, exerting protective effects against non-communicable diseases (NCD) including cancer and cardiovascular (CVDs), metabolic, and neurodegenerative disorders (NDDs). These benefits have been associated with the presence of secondary metabolites, namely polyphenols, glucosinolates, carotenoids, terpenoids, alkaloids, saponins, vitamins, and fibres, among others, derived from their antioxidant, antiatherogenic, anti-inflammatory, antimicrobial, antithrombotic, cardioprotective, and vasodilator properties. Polyphenols as one of the most abundant classes of bioactive compounds present in plant-based foods emerge as a promising approach for the development of efficacious preventive agents against NCDs with reduced side effects. The aim of this review is to present comprehensive and deep insights into the potential of polyphenols, from their chemical structure classification and biosynthesis to preventive effects on NCDs, namely cancer, CVDs, and NDDS. The challenge of polyphenols bioavailability and bioaccessibility will be explored in addition to useful industrial and environmental applications. Advanced and emerging extraction techniques will be highlighted and the high-resolution analytical techniques used for FBCs characterization, identification, and quantification will be considered.

## 1. Introduction

Non-communicable diseases (NCDs), mainly cardiovascular diseases (CVDs), cancer, chronic respiratory diseases, neurodegenerative diseases (NDs), and diabetes, represent a severe burden worldwide. According to the most recent data, NCDs caused over 70% of the 57 million deaths worldwide in 2016 [1]. The major risk factors contributing to this scenario have been identified as the combination of an unhealthy diet, poor physical activity, and alcohol and smoking abuse. Diet, in particular, is considered a leading risk factor for illness, death, and disability and it is estimated that one in five deaths are associated with a poor diet [2]. For this reason, dietary intervention holds great potential in the prevention, mitigation, or even treatment of the most prevalent NCDs. In fact, epidemiological evidence shows that the prevalence of CVDs, cancer, diabetes, and neurodegenerative diseases is lower in populations with diets involving higher ingestion of bioactive compounds, namely the Mediterranean diet [3,4]. For this reason, “functional foods” containing components that can protect our health are becoming very popular. In this context, this review is devoted to the effect of bioactive compounds present in food—in particular, phenolic compounds. Beyond their characterization, we will span the protective effects of polyphenols in cancer, CVDs, and NDDs; highlight pertinent questions regarding their bioavailability and bioaccessibility; and explore new applications in environment, food, and cosmetic industries. Finally, emerging sustainable extraction techniques and methodologies for polyphenols quantification will also be addressed.

## 2. Food Bioactive Compounds from Triterpenes to Polyphenols

Food bioactive compounds (FBCs) refer to all compounds, which are mostly without nutritional value and naturally present in food that exert a certain bioactive effect on the human body [5,6,7]. Often, such biological activity is acknowledged in compounds that have a positive effect, but this classification is very narrow because negative effects on an organism are also a form of bioactivity. This includes adverse effects such as toxicity, allergenicity, and mutagenicity, which are usually dependent on the dose and bioavailability of a given substance [5]. Nevertheless, in this review, FBCs will designate the dietary compounds that can elicit protective effects in our organism, therefore promoting its health and fitness. In this context, bioactive compounds present in popular beverages, such as tea, coffee, and wine, that constitute important sources of these compounds in the human diet will also be considered. FBCs encompass a wide range of compounds, mostly produced by plants as secondary metabolites, and used in different functions as competition, defence, attraction, and signalling [6,8] (Figure 1). Flavonoids, for instance, are protective agents against free radicals generated during photosynthesis. In turn, some terpenoids have been reported to attract pollinators or seed dispersers or inhibit competing plants and different alkaloids are used to repel herbivores and insects [6]. FBCs can be essentially found in fruits, vegetables, grains, and leaves and despite their importance for our health, there are no recommended daily intake values for the ingestion of most of the FBCs as it happens with proteins, lipids, or vitamins [5,9].

### 2.1. FBCs Classification

The systematization and characterization of the myriad of FBCs reported so far, certainly above several dozens, constitutes a huge challenge that is being addressed by different researchers worldwide. Regarding this, three main databases are aggregating the data reported in the literature on the bioactive components of different foods: the USDA flavonoid database [10]; Phenol-Explorer [11] covering polyphenols; and eBASIS (Bioactive Substances in Food Information Systems) [7]. The eBASIS is an EU-funded multinational platform that has a broader focus and includes curated information about the composition and beneficial effects of additional bioactive compound classes beyond polyphenols, such as glucosinolates, phytosterols, and capsaicins [7]. Currently, eBASIS deploys information of over 300 foods and 17 compounds classes, including for the first time in this type of platform, the bioactive composition of meat [7]. In fact, the existence of FBCs is not exclusive to plants and they can be present in foods from animal origin, such as meat and dairy (reviewed in [12] and [13]). Fungi, notably edible and wild mushrooms, have been unveiled as suitable sources of non-plant FBCs. The structural diversity and effects that, for instance, wild mushrooms’ FBCs present are still poorly studied. Nevertheless, many phenolic compounds, tocopherols, ascorbic acid, and carotenoids have already been reported in the composition of these fungi (reviewed in [14]). Fermented foods are very popular in the human diet because of numerous recognized bioactive effects, namely antihypertensive and antioxidant properties. These effects are mostly elicited by FBCs (probiotics) produced by the microorganisms responsible for the fermentation process, usually yeasts and bacteria [15].

As already referred to, the range of FBCs available in nature is so diverse in terms of origin, structure, and bioactive effects, therefore it would be impossible to explore all in detail. Therefore, in this review, we will refer very briefly to the most abundant FBCs and focus our attention on polyphenols in the following sections.

### 2.2. Overview of Main FBC Classes

As condensed in Figure 1, the major FBCs are those produced by plants and can be grouped as polyphenols, triterpenes and phytosterols, terpenoids, polysaccharides, capsaicinoids, carotenoids and tocopherols, alkaloids and glucosinolates, among others. Triterpenes and their functionalized forms (triterpenoids) encompass at least 18 subclasses of compounds, from which squalene derivatives and saponins are the most well-known. Among such a wide diversity, lupane, oleanane, and ursane seem to be very promising molecules as anti-cancer agents (reviewed in [16]). Phytosterols are also derived from triterpenes and they are key components of all plant cells’ structure, having a similar role to cholesterol in animal cells. Remarkably, several clinical studies showed that a diet rich in phytosterols, such as campesterol, sitosterol, and stigmasterol (Figure 1), promote a relevant lowering effect of low-density lipoprotein (LDL) cholesterol (reviewed in [17]). Terpenoids, also known as isoprenoids, constitute the largest group of secondary metabolites, with tens of dozens of compounds, making them ubiquitous to all living cells and organisms. Terpenoids are particularly abundant in plants and fruits, with many of them being volatile at room temperature and responsible for the characteristic aroma of plants and fruits, such as limonene in several citrus fruits (Figure 1) (reviewed in [18]). Carotenoids (as β-carotene) and tocopherols (as α- and γ-tocopherols) were grouped as FBCs as they are the direct precursors of vitamin A and E, respectively, which are essential components of the human diet. Alkaloids also constitute a broad group of compounds exhibiting alkali-like properties and at least one nitrogen atom in a heterocyclic ring structure [19]. Caffeine, which is an alkaloid, is one of the most popular FBCs in the human diet. It is mostly obtained from coffee and tea. Glucosinolates (GSLs) are anions with a thiohydroximate-O-sulfonate core group linked to glucose and a highly variable side chain (reviewed in [20]). GSLs are mainly found in cruciferous plants typical in the Mediterranean diet (broccoli, cabbage, cauliflower, brussels sprouts) and are responsible for their specific taste and odour. Many GSLs have been associated with different bioactivities in our organism, including chemopreventive and anti-inflammatory effects. This is the case of sulforaphane, a compound that exists in broccoli and is being explored as a promising anticancer agent. Eventually, this type of compounds may explain the epidemiological results correlating the consumption of cruciferous vegetables with a lower oncological incidence (reviewed in [20]). Polysaccharides designate the polymeric chains of monosaccharides linked by glycosidic bonds ubiquitously found in nature. Starch and glycogen (energy reservoir in plant and animal cells, respectively) and cellulose (structural backbone in vegetal cells) are examples of well-known polysaccharides, but there are many other FBCs within this class of compounds that are very relevant to human health (reviewed in [21]). Among them, inulin, a dietary fibre, and its derivatives fructooligosaccharides (FOS) were shown to have an important contribution to the homeostasis of host–gut microbiota symbiosis and consequent overall well-being. Such effects seem to positively affect infant nutrition, blood sugar, and lipid metabolism, mitigating the risk for obesity and colon cancer development (reviewed in [21]). Polyunsaturated fatty acids (PUFAs), like linoleic acid, eicosapentaenoic acid (EPA), and docosahexaenoic acid (DHA), also constitute important FBCs. Such relevance is based in the epidemiological correlation between a diet poor in certain PUFAs, particularly those from the ω-6 and ω-3 families, and the high prevalence of chronic diseases such as cancer, CVDs, and diabetes. This correlation seems to involve the modulation of the inflammatory response (at the level of gene expression and cell membrane composition) elicited by lipid mediators derived from ω-6 and ω-3 PUFAs. Nevertheless, overconsumption of these compounds should be avoided because it can trigger oxidative damage to cells’ structures, particularly to the blood vessels walls, which is a hallmark in cardiovascular diseases progression (reviewed in [22]). As mammals are unable to synthesise PUFAs, they must be obtained from the diet. Luckily, there are many PUFAs sources available, such as walnuts, seeds, and oils from sunflower, flax, soybean and corn, or fishes such as salmon, mackerel, and herring, among others. Furthermore, metabolic engineering is being used to increase PUFAs production, namely arachidonic acid, in different plants and organisms used in our diet [23]. There are many peptides encrypted in the structure of the dietary protein that upon proteolytic cleavage originate molecules such as carnosine, creatine, taurine, or carnitine, just to name a few examples [13]. These peptides have been identified in different meats, such as pork, beef, or chicken, and various species of fish and marine organisms and they are considered FBCs due to several bioactive effects that they elicit in our organism. This includes, among others, mitigation of muscle wasting diseases (such as sarcopenia), decreased risk of metabolic syndrome (via reduction of food and caloric intake), blood pressure homeostasis (via ACE-inhibitory components from connective tissue), and preservation of a functional gut environment (through the supply of meat-derived nucleotides and nucleosides) (reviewed in [14]).

## 3. Structure, Classification, and Biosynthesis of Polyphenols

### Structure and Classification

Polyphenols constitute one of the most common and widespread groups of phytochemicals in the plant kingdom with more than 8000 phenolic structures currently known [24]. This heterogeneous group of compounds is chemically characterized by an aromatic ring with at least one hydroxyl group and their structure can vary from simple molecules, like phenolic acids, to highly polymerized compounds, such as tannins [25,26].

There are several ways to classify polyphenols including their origin, biological function, and chemical structure [24,27]. This last one is probably the most adopted classification and divides polyphenols into two main groups: flavonoids and non-flavonoids (Figure 2) [25].

Flavonoids are very abundant in plant-based foods, such as fruits, vegetables, chocolate, tea, wine, among others, and to date, several hundred different flavonoids have been described and the number continues to increase [28,29,30]. In general, these compounds share a common basic structure of diphenylpropanes (C6-C3-C6), which consists of two benzene rings (Rings A and B) connected by a 3-carbon bridge, which usually forms an oxygenated heterocycle (Ring C) [31]. Based on the hydroxylation pattern and variations in the heterocyclic ring, flavonoids can be divided into six major subclasses including flavonols, flavones, isoflavones, flavanols, anthocyanidins, and flavanones [25].

Flavonols are mainly characterized by the presence of unsaturation in the heterocyclic ring, between the C2 and C3 carbons, a hydroxyl group in position 3, and by the presence of a ketone group in C4 [32]. Quercetin and kaempferol are the predominant phenolic compounds in this class followed by myricetin, isorhamnetin, fisetin, morin, and their glycoside derivatives [33]. Flavones differ from flavonols in terms of the absence of the 3-hydroxyl group in Ring C and the main components of this group are apigenin and luteolin. They are mainly present in their glycoside forms [25].

Flavanols or flavan-3-ols are another important class of flavonoids that are characterized by the presence of a saturated heterocyclic ring and a hydroxyl group in C3 [32]. Flavanols exist in both the monomer form (catechins) and the polymer form (proanthocyanidins, which are traditionally considered to be condensed tannins) [24,27]. Catechin and epicatechin are the predominant flavanols in fruits, while epigallocatechin, gallocatechin, and epigallocatechin gallate are commonly found in seeds of leguminous plants, grapes, and in tea [34,35].

Flavanones have a saturated heterocyclic ring, like flavanols, but with a ketone group in C4 and no hydroxyl group in C3. These flavonoids constitute a minority group in food, despite the fact they appear at high concentrations in citrus and tomatoes, and in certain aromatic plants [27,36]. The main aglycones are eriodictyol found in lemons, hesperedin in oranges, and naringenin in grapefruit [36].

Isoflavonoids, such as isoflavones, differ from the other classes of flavonoids because the Ring B is attached to C3 in Ring C instead of C2. Isoflavones have structural similarities to estrogens and they are occasionally referred to as “phytoestrogens” [37]. These flavonoids are found almost exclusively in leguminous plants, in which daidzein and genistein are the main isoflavones in soy [38].

Anthocyanidins are another important group of flavonoids that are responsible for the red, blue, and purple colours of the majority of flowers, fruits, vegetables, and certain varieties of grains, such as black rice [39]. In terms of chemical structure, anthocyanidins are polymethoxy or polyhydroxy derivatives of the flavylium cation, containing a C15 backbone structure, and the most commonly found are cyanidin, delphinidin, pelargonidin, peonidin, petunidin, and malvidin [40]. When these flavonoids are found in their glycosidic form, i.e., linked to a sugar moiety, they are called anthocyanins [32]. Commonly, the sugars bonded to anthocyanidins are monosaccharides (galactose, glucose, arabinose, and rhamnose) and di- or tri-saccharides formed by the combination of the four monosaccharides [41]. The main differences between anthocyanins and anthocyanidins results from (i) the number, the location, and the nature of sugars bonded to the molecule; (ii) the number and type of aromatic or aliphatic acids linked to the sugar; (iii) the position and number of hydroxyl groups; and (iv) the degree of methylation of these groups [42].

Regarding non-flavonoids compounds, this group includes phenolic acids, stilbenes, lignans, and tannins. Phenolic acids are compounds characterized by a phenol ring that contain at least one carboxylic acid functionality [43]. In general, they are derived from two main phenolic compounds, namely benzoic acid and cinnamic acid. These compounds occur predominantly as hydroxybenzoic and hydroxycinnamic acids and may be found either in their free or conjugated forms [44]. The hydroxybenzoic acids are the simplest phenolic acids found in nature and contain seven carbon atoms (C6-C1), while hydroxycinnamic acids are characterized by a C6-C3 chain and are rarely found in their free form in plants [27]. Typically, they exist in the form of esters of hydroxy acids, i.e., quinic, shikimic, and tartaric, besides they are derivatives of sugars [32]. Caffeic, *p*-coumaric, ferulic, and sinapic acids are some representative examples of hydroxycinnamic derivatives, while vanillic, gallic, and syringic acids belong to hydroxybenzoic acids.

Stilbenes are a group of polyphenols derived from the phenylpropanoid pathway, which are characterized by two phenyl rings connected by a two-carbon methylene bridge (C6-C2-C6) [45]. Among stilbenes, resveratrol is probably the most prominent compound and one of the most studied due to its potent biological activities [46]. It has been found in several foods including grapes, peanuts, mulberries, and red wine [46].

Lignans is a group of natural non-flavonoids phytoestrogens structurally characterized by the combination of two phenylpropanoid units connected at the β and β’ carbon atoms [47]. They have a wide occurrence in nature and the main sources of dietary lignans are whole-grain cereals, oilseeds, fruit, vegetables, and some beverages, such as tea, coffee, and wine [25]. These compounds may occur in the form of aglycones and glycosides and the most predominant dietary lignans are secoisolariciresinol, matairesinol, medioresinol, pinoresinol, lariciresinol, and syringaresinol [25].

Tannins are another large group of complex biomolecules of phenolic nature synthesized by a wide diversity of plants [48]. Based on their chemical structure, tannins are divided into four main categories: hydrolysable tannins, which are further subdivided into gallotannins and ellagitannins, condensed tannins, phlorotannins, and complex tannins. Gallotannins are tannins constituted by galloyl units or their *meta*-depsidic derivatives that are linked to diverse polyol-, catechin-, or triterpenoid units, while ellagitannins are tannins with at least two C–C coupled galloyl units with no glycosidically-bonded catechin unit [49]. Condensed tannins, also called proanthocyanindins, are oligomers or polymers composed of flavan-3-ol nuclei [50]. The most common basic units of condensed tannins are (+)-catechin, (+)-gallocatechin, (−)-epicatechin, (−)-epigallocatechin, and (−)-epigallocatechin gallate [50].

Phlorotannins, of which their common structural base is the phloroglucinol, are a class of tannins found in brown algae (Phaeophyta) [48]. Examples of phlorotannins identified in marine brown algae are phloroglucinol, eckol, fucodiphloroethol G, phlorofucofuroeckol A, 7-phloroeckol, dieckol, and 6,6-bieckol [51].

Complex tannins are a particular group of tannins in which a catechin unit is linked glycosidically to an ellagitannin or a gallotannin unit [49]. As the name implies, the structure of these compounds can be very complex, as in the case of Acutissimin A and eugenigrandin A, in which a gallocatechin or catechin, respectively, are complexed through a carbon–carbon bond to an ellagitannin unit [48].

The biosynthesis of polyphenols involves a complex network of routes that include the metabolism pathways of shikimic acid and malonic acid [52]. While the first one produces the most plant phenolic compounds, the second pathway (malonic acid) is an important source of phenolics in bacteria and fungi but is less significant in higher plants [52]. The shikimate pathway converts simple carbohydrates derived from the pentose phosphate pathway and glycolysis into the aromatic amino acids tyrosine and phenylalanine, which are the precursor compounds of the phenylpropanoids (Figure 3) [53]. In most vascular plants, phenylalanine is the primary substrate for phenylpropanoid synthesis, while tyrosine is used to a lesser extent in some plants [53].

The key reaction that connects primary to secondary phenolic metabolism is catalysed by the enzyme phenylalanine ammonia lyase (PAL), where L-phenylalanine is deaminated to produce *trans*-cinnamic acid, leading to the C_6_–C_3_ structures [52]. The final intermediate 4-coumaroyl-CoA and three molecules of malonyl-CoA are then condensed to produce naringenin chalcone by the enzyme chalcone synthase (CHS) [24]. Chalcone is isomerized by chalcone flavanone isomerase into a flavanone, which is the basic skeleton to all flavonoid classes [24].

## 4. Polyphenols and Health

A substantial body of evidence that has accumulated over the past decades indicates that phytochemicals, including polyphenols, terpenoids, alkaloids, and sulphur-holding compounds, can have positive outcomes on human health by virtue of their biological activities, such as antioxidant, antimicrobial, and anti-inflammatory activities. Amidst phytocomponents, polyphenols are the most investigated worldwide. Contemporary research has demonstrated that the mechanism of action implied in the protective effects of polyphenols occurs via cellular signalling pathways and it is not directly triggered by epigenetic actions in the context of physiological and pathological limitations. Besides, recent studies have evidenced that the polyphenol ingestion of more than 1 g/day is correlated with diminished onset and progress of chronic ailments related to oxidative stress (Figure 4), especially CVDs and NDs, age-related pathologies, type 2 diabetes mellitus, as well as several forms of cancer [56,57].

### 4.1. Cancer

Cancer is a disease induced by the abnormal and uncontrolled growth of cells that can interfere with the normal functions of the host. Worldwide, one in six deaths is caused by cancer, which corresponds to 30% of all premature deaths in adults (30–69 years). Furthermore, the number of cancer deaths is expected to double in the current century [58]. It is established that cancer development may be influenced by hereditary factors and/or lifestyle. In fact, the pathways that lead to cell mutagenesis remain poorly elucidated. However, it is known that the components of diet can exert a protective action against carcinogenesis [58,59]. Polyphenols are natural compounds present in many plants that according to literature, influence in the prevention of malignancy neoplasia [58,59]. The formation of reactive oxygen species (ROS) is reputable to produce several negative outcomes in the organism, such as inflammatory and chronic diseases. Under normal conditions, ROS production is balanced by cellular antioxidant activity; however, it can be affected by immune responses against extracellular pathogens and inflammation, which can lead to overproduction of ROS, resulting in an imbalance between pro-oxidant and antioxidant systems. The excess of ROS may be a factor that increases cell proliferation through mutations in DNA, leading to the development of carcinomas [59,60]. Thus, the antioxidant activity of the polyphenols can favour the normal functioning of cells. Also, some compounds, called anti-cancer, act on genome stability, and increase the cellular antioxidant protection by induction of antioxidant enzymes [59,61]. Some polyphenols, such as resveratrol, quercetin, apigenin, curcumin, among others, act in the upregulation of p53 protein expression in several cancer cell lines, through phosphorylation, acetylation, and reduction of oxidative stress, which can lead to cell cycle arrest, DNA repair, and finally, to apoptosis of malignant cells. Interestingly, the increasing p53 expression can overcome chemoresistance of cancer cells by reduction of mutation p53 [60]. Table 1 brings a compilation of recent studies that evidence the potential of polyphenols as therapeutic agents against several types of cancer. Some authors verified that low dose concentrations of polyphenols have decreased cell cancer viability in in vitro models. For instance, the flavones hispidulin and luteolin showed high cytotoxic actions against human leukaemia cells (MV4-11 cell line), being the induced cell apoptosis achieved at lower concentrations, with IC_50_ values of 8.2 and 3.1 μM, respectively [62]. Overall, in in vivo studies, these bioactive compounds act mainly by inhibiting tumour growth and reducing their size. Additionally, some flavonoids have displayed inhibitory effects against cancer metastasis, as in the case of anthocyanins and resveratrol, which prevented melanoma lung metastasis in mouse models [63,64].

Polyphenols have shown antiproliferative activities on multidrug-resistant cells. This is the case for alpinumisoflavone, which in low concentrations (IC_50_ value of 5.41 μM), inhibited the growth of doxorubicin-resistant leukaemia cells (CEM/ADR5000 cell line) [65]. Similarly, quercetin has the ability to reverse the resistance of lung cancer cells to paclitaxel drug, inhibiting Akt and depolarization of mitochondrial membrane potential, while exerting synergistic effects with the cancer-drug, increasing its therapeutic action [66]. Luteolin has also presented a synergetic effect in lung cancer treatment with tumour necrosis factor-related apoptosis-inducing ligand (TRAIL) [67]. Moreover, the combined use of polyphenols with chemotherapy can be interesting to mitigate the side effects of treatment, such as reported in the work of Tajaldini et al. [68], where they verified that the treatment against oesophageal cancer in mice using doxorubicin combined with naringin or orange-peel extract inhibited tumour growth, successfully increased apoptotic cell death, and in contrast to the chemotherapy used alone, resulted in lower systemic toxicity, decreased oxidative stress, and body weight maintenance.

### 4.2. Cardiovascular Diseases (CVDs)

CVDs are the main causes of death worldwide. The incidence of these disorders increases with non-modifiable factors, e.g., age, gender, and hereditary conditions; however, modified factors also have an important influence in the prevention or development of CVDs, such as cholesterol levels, obesity, hypertension, type 2 diabetes mellitus control, diet, smoking, stress, and other conditions [61,69]. Some polyphenols have been extensively investigated due to their cardioprotective actions, namely vasodilator and antiplatelet potentials, as well as their ability to reduce both blood pressure and LDL-cholesterol, as presented in Table 2. Hypercholesterolemia is a condition that significantly increases the risk of CVDs, mainly atherosclerosis [70]. Numerous studies have verified the positive effects of polyphenols ingestion on blood cholesterol levels. In clinical trial studies, daily anthocyanin supplementation (320 mg/24 days) played a significant role in the reduction of total cholesterol (TC) and LDL-cholesterol (LDL-C) [70]; oral administration of catechins from green tea (1315 mg/day/12 months) was efficient to reduce LDL-C in postmenopausal women without altering high-density lipoprotein cholesterol (HDL-C) concentrations [71].

In diabetic patients, treatment with polyphenols also proved to be effective in controlling the parameters of the disease. Resveratrol treatment (800 mg/day/2 months) has shown promising results in type 2 diabetes patients: it not only improved antioxidant activity in the blood, but also reduced both body weight and blood pressure [80]. In a diabetic-mouse model, this stilbene administration lowered blood glucose, plasma triglyceride levels, and inflammation factors [81].

Some authors report the protective effects of curcumin in patients with CVDs. This curcuminoid has high anti-inflammatory and antioxidant activities that act on the factors of transcription regulation, which in turn make them act against the development and progression of CVDs, such as atherosclerosis, myocardial infarction, stroke, cardiac hypertrophy, and aortic aneurysm [69].

Overweight and obesity are conditions associated with several CVDs. In some cases, an imbalance in the individual’s intestinal flora may be a highly contributing factor to these conditions. Regarding this, in vivo studies have shown that the oral administration of polyphenols, namely catechins, anthocyanins, and procyanidins, simulate the growth of prebiotics microorganisms, e.g., *Lactobacillus, Bifidobacterium, Akkermansia, Roseburia*, and *Faecalibacterium* spp, which lead to a reduction of fat mass, liver steatosis, body weight gain, and also to the regularization of glucose metabolism [90].

### 4.3. Neurodegenerative Diseases (NDDs)

The population’s life expectancy has increased worldwide, which may determine the rise of the burden of age-related global diseases, mainly neurodegenerative diseases (NDDs), since the occurrence of such disorders is associated with age progression (prevalence tends to increase after 65 years). When the human central nervous system (CNS) stops functioning properly, neuron regeneration is inhibited, thus leading to death. A significant increase in neuronal loss happens in Alzheimer’s disease (AD), Parkinson’s disease (PD), and multiple sclerosis, among others. Although NDDs have been widely studied, so far, there is no cure for such deleterious illnesses. NDDs treatments aim to slow down the degenerative processes of neurons [91]. Oxidative stress has been addressed as a major contributor to the development and progression of neurodegeneration; thus, several studies have explored the use of polyphenols against NDDs, as presented in Table 3. A diet rich in polyphenols can contribute to the neurological health of the elderly. For example, the daily ingestion of anthocyanin-rich cherry juice (200 mL/12 weeks) by older adults (>70 years) with mild to moderate dementia during 12 weeks was sufficient to significantly improve the verbal fluency, as well as short- and long-term memory [92]. In addition, the oral administration of polyphenols from white grape juice (20–40 mg/kg/day) decreased inflammatory and oxidative processes in autoimmune encephalomyelitis-induced mice, which is the most used model for multiple sclerosis in vivo [91]. AD is a chronic progressive neurodegenerative disorder resulting in disturbances of memory and cognitive function, which is associated with neuroinflammation and deposition of amyloid- beta (Aβ) in the brain. Extracts of *Arabidopsis thaliana,* rich in polyphenols, namely derivatives of the flavonols quercetin and kaempferol, have mitigated the neuroinflammation induced by Aβ through activation of the nuclear factor erythroid 2-related factor 2 (Nrf2)-antioxidant response. In an in vivo assay, the same extract efficiently restored the locomotor activity of AD *Drosophila melanogaster* flies expressing human Aβ1–42 [93]. Withal, polyphenols have displayed protective activity against the side effects of chemotherapy in neurological cells, a disorder known as chemobrain, which can affect > 75% of cancer patients [94]. According to Shi et al. [94], resveratrol administered to mice (100 mg/kg/day/3 weeks) exerted neuroprotective effects against chemobrain induced by cancer treatment with docetaxel, adriamycin, and cyclophosphamide (DAC).

## 5. Bioaccessibility and Bioavailability of Polyphenols

Polyphenols are present in several components of a normal diet and the intake of these compounds can be fundamental for maintaining good health. However, the processes of metabolization, transport, and distribution of these compounds to the target organs are very complex and can affect the structure and, consequently, the bioactivities of such compounds [100,101]. Absorption of polyphenols in the gastrointestinal tract (GI) depends mainly on two factors: bioaccessibility and bioavailability. Bioaccessibility is associated with the amount of these compounds that is available to undergo the process of metabolization; it may be influenced by the interaction of polyphenols with food components, e.g., proteins, lipids, and carbohydrates, and by food [100,101], while bioavailability refers to the ability of these compounds to be metabolized and distributed throughout the body (Figure 5).

In general, polyphenols have poor bioavailability since many factors limit their metabolism, such as solubility, the complexity of the chemical structure, degree of polymerization, interaction with other molecules, among others [100,103]. McClements [103] describes the bioavailability (BA) of a phytochemical as a variable dependent on the four fundamental factors: BA = S* × B* × T* × A*, where S* corresponds to the stability of the compound after food processing and/or storage conditions; B* is the bioaccessibility (B*); T* is the fraction of compounds that remains intact after passing through the GI to the absorption site; and A* is the fraction of compounds that is actually absorbed by the epithelium cells. Phenolics metabolization starts in the oral cavity, where glycosidase enzymes act on the linking of glycosylated molecules; in the stomach, a greater release of polyphenols from the food matrix occurs and some compounds may be hydrolysed due to the acid medium (pH 2–4) [100,101,103]. After this, the main biotransformation of these compounds occurs in the gastrointestinal tract (GI), determining their bioavailability prior to absorption and circulation in blood vessels [100,102]. Metabolization in GI is divided into phase I and II, which are carried out in gut and liver cells. Degradation by microflora can also occur at the colon level. In phase I, phenolic compounds undergo oxidation, reduction, and hydrolysis, which are responsible for changes in their structure, including amino, carboxyl, and hydroxyl groups [100,101,103]; whereas in phase II, enzymatic reactions are responsible for reducing the toxicity of the compounds and elimination via glucuronidation, sulfation, and methylation [100,102].

It is estimated that between 5–10% of polyphenols from the food matrix are actually absorbed and transported to the liver [101]. The pH changes from gastric (pH 2) to intestinal phase (pH 6–7.5) influence flavonoid bioaccessibility since most of them are resistant to acid hydrolysis but are depredated in the alkaline condition [100]. Therefore, it is quite clear that expressing the amount of polyphenols ingested to exert their bioactivities in vivo is a complex process, which goes beyond the relationship between polyphenols ingested and eliminated in the urine as after several reactions, these compounds are converted into other metabolites or degraded along the way [100,102].

To overcome the barriers of human metabolism and increase the absorption of phenolic compounds, some strategies are being developed, such as the application of food processing to increase their bioavailability [102], the use of food matrices that have an effect on the protection of the phenolic compounds against degradation [100,103], and the development of delivery systems that exert protection on the molecules till they reach the target site where they must exert their bioactivities [103]. For instance, the freezing process was able to increase the bioaccessibility of strawberry flavonoids (64.4 → 90.8%) and anthocyanins (47.2 → 83.4%) [104]. Additionally, the total content of anthocyanins from highbush blueberry purée in the human blood was improved after a blanching treatment was applied to this food product [105]. The stability of flavonoids tends to increase when they are incorporated into food matrices containing milk proteins (McClements, 2020). Green tea polyphenols added to cheese were released gradually from the tea-intra cheese in the gastrointestinal environment, which lowered their degradation rate and enhanced their antioxidant activity [106]. Similar effects were verified with anthocyanins incorporated into diverse food matrices, e.g., milkshake and omelette, where their stability was increased after oral phase, gastric, and intestinal digestions [107].

To enable polyphenols applications as therapeutic agents, the use of delivery systems seems to be one of the most promising strategies as these techniques can increase the stability of these biomolecules under gastrointestinal conditions and may also facilitate their absorption by the epithelium cells [103,108]. Nanoencapsulation with whey protein, casein, and Zein protein have shown great potential to increase the bioaccessibility of green tea flavan-3-ols and anthocyanins, among others [101]. Lipid-based nanoencapsulations have increased bioavailability of anti-cancer polyphenols, e.g., resveratrol, curcumin, genistein, and quercetin. This type of delivery system has the main advantage of increasing the solubility of hydrophobic compounds in GI [103,108]. According to the review carried out by Lagoa et al. [108], several in vitro and in vivo studies show that delivery systems, such as liposomes, nanoemulsions, and polymeric/biopolymeric nanoparticles, potentiate the effects of polyphenols in targeted therapies for several cancer types.

## 6. Potential Applications—Food and Cosmetic Industries and Environment

In addition to their broad use as alternative therapeutic agents, polyphenols can be explored in several technological fields for the development of new products and/or technologies (Figure 6).The main interest of such applications is the employment of natural and less aggressive compounds to human health and to the environment [100]. In this way, several polyphenols have been investigated for their ability to act in the treatment of polluted waters. This is the case of tannins, which due to their ability to complex with proteins and metal ions, have been used as biosorbents in contaminated water with heavy metals, dyes, surfactants, and pharmaceutical compounds [100,109]. Tannin-based water treatment products are already commercialized, such as TaFloc^®^, a tannin coagulant from TANAC S. A. (www.tanac.com.br).

The textile industry is one of the most polluting, producing many chemicals, like artificial dying-loaded wastewaters. Polyphenols possess a range of colours that vary from yellow to purple and for this reason, they have been exploited in the development of natural dyes with less toxicity than the conventional ones used in silk and wool fabric dyeing [100]. Moreover, natural dyes obtained from oak barks (rich in gallotannin, ellagitannin, and quercetin) exerted UV protection and antimicrobial activity when applied to Tussah silk [110]. UV protection was also achieved by dyeing cotton with phenolic tea extracts [111]. Natural dyes based on tannins (e.g., from acacia extract, oak galls extract, quebracho extract, and others) and based on anthocyanins (e.g., from sorghum extract), are widely marketed by eCommerce suppliers (e.g., http://www.wildcolours.co.uk).

Since the changes in the consumer’s perspective in view of the controversy over synthetic additives has substantially increased the choice for more natural products, polyphenols have become molecules of high interest in the food industry. The preference for natural additives can be a strategy for the development of innovative and more beneficial health food products, e.g., functional foods. Thus, the bioactivities of polyphenols have been investigated for their application as food preservatives. The addition of lychee pericarp extract in sheep meat nuggets (1.5%) showed prevention against lipid oxidation similar to butylated hydroxytoluene (BHT, 100 ppm), a common synthetic antioxidant [112]. A similar result was also observed with the use of an ethanolic extract of rice (2%), rich in phenolic acids, in mayonnaise, where the effect of the natural extract was statistically equal to that of the synthetic additive (0.5%). Furthermore, a natural additive displayed antibacterial action against *Staphylococcus aureus,* coliforms, and mesophyll bacteria and also inhibited the growth of moulds and yeasts [113]. Polyphenols, especially anthocyanins, are being used by the food industry as a colouring additive, under the additive-number E163 [114]. Anthocyanins isolated from flowers and berries residues have provided stable red-purple colour when employed as natural colourants in different food matrices [114,115]. Moreover, anthocyanin increased antioxidant activity and decreased microbiological degradation in fresh sausage [115]. These examples clearly show that polyphenols can be incorporated into different food systems, allowing a nutritional enrichment with bioactive compounds and producing functional products with added value to human health protection [61].

In another segment of the food industry, phenolic compounds are used as components of smart/bioactive packaging [116,117,118]. For example, anthocyanins have been used as indicators of fish freshness [117] and as anti-lipidic peroxidation agents in olive oil packing [116]. Also, tannin-protein-based edible films have shown antioxidant activity and antimicrobial activity against *Escherichia coli* and *Listeria innocua* [118].

Following the trend of replacing synthetic substances with natural compounds, the cosmetic industry has also been exploiting phytochemicals [100]. As is well known, polyphenols hold sundry biological potentialities, such as in vivo and in vitro skin protection and anti-ageing activities. In a clinical trial, an O/W emulsion enriched with a phenolic extract from *Nymphaea rubra* significantly increased skin elasticity, also reducing skin roughness and wrinkles after 60 days of use. Such positive effects were correlated with antioxidant and anti-tyrosine activities [119]. Phenolic acids, namely *p*-hydroxybenzoic, *p*-coumaric, and protocatechuic acids, showed stable bioactivities, e.g., antioxidant, anti-tyrosinase, and anti-inflammatory effects, when added in a cosmetic cream base [120].

Some polyphenols have chromophore molecules in their structure, which allow them to absorb ultraviolet radiation, thus blocking the entry of solar radiation into the skin [100]. In a mouse model, the oral administration of caffeic acid (100 mg/day/8 days) inhibited dermatitis and melanogenesis caused by UVB irradiation exposure [121]. A *Nephelium lappaceum* peel extract, rich in tannins and flavonoids, was able to improve the sun protection factor (SPF) of a cream base emulsion containing ethylhexyl metoxycinnamate, allowing the reduction of the synthetic anti-UV filter concentration in the formula, which consequently minimized both the toxicity of the product and its production costs [122].

Regarding the hair care niche, epigallocatechin-3-gallate from green tea displayed great outcomes on human hair growth due to its proliferative and anti-apoptotic effects on dermal papilla cells [123]. Likewise, a procyanidinic extract of *Malus pumila Miller* cv Annurca stimulated the hair growth-promoting activity after chemotherapy-induced alopecia [124]. Chemotherapy can also induce nail damage, causing disfigurement and pain. The use of a polyphenolic-rich balm (a blend of natural oils and waxes) not only prevented nail damage, but also secondary infections in a trial with 60 cancer patients [125]. Currently, this medicinal cosmetic is available on the market under the brand Polybalm^®^—polyphenol-rich nail remedy (http://polybalm.com/).

## 7. Emerging Extraction Techniques and Methodologies for Polyphenols Quantification

The extraction procedure is a very important step with great impact on the analysis and characterization of bioactive compounds. Considering their low concentrations in foods and other natural sources, the use of extraction procedures that efficiently allows their recovery from bulk matrices and simultaneously removes some potential compounds that can interfere in the analytical procedure is of utmost importance.

The general workflow followed in food analysis is based on four important steps (Figure 7): (i) sample preparation, which includes the sample pre-treatment and/or treatment, the extraction procedure, the extract clean up, and in most cases, extract concentration by evaporation; (ii) the data acquisition using high-resolution analytical techniques, namely gas chromatography mass/spectrometry (GC-MS), liquid chromatography/mass spectrometry (LC-MS), and nuclear magnetic resonance (NMR), depending on the aim of the study and targeted analytes; (iii) the data acquisition processed by the equipment software; and (iv) multivariate statistical analysis, which may include analysis of variance (ANOVA), principal components analysis (PCA), the partial linear square analysis combined with discriminant analysis and cross-validation, among others, depending on the aim of the study.

Typically, the sample pre-treatment/treatment is the most time-consuming step. It depends on sample complexity, physical state of the sample-liquid or solid, and the specific analytical needs (target analytes). Filtration, dilution, and centrifugation are procedures most related with liquid samples, whereas solid samples are usually submitted to drying (freezing with liquid nitrogen, freeze-drying, or air drying) followed by a milling or grinding process to reduce the particle size. This is important to increase the contact surface and consequently improve the extraction efficiency. The ideal procedure for sample preparation should remove potential interferents, present high recoveries and high reproducibility, be easy to automate for high sample throughput, be rugged, agree with green chemistry principles, and very importantly, should be fast and cost-effective. Figure 8 present a scheme of the strategy usually employed for the sample pre-treatment before analysis.

There has been unprecedented growth in measurement techniques over the last few decades. High-resolution analytical techniques, such as chromatography, spectroscopy, and microscopy, as well as sensors and microdevices, have undergone phenomenal developments, which make bioactive compound analysis easier than before. However, sample preparation is often the bottleneck in a measurement process. Indeed, the most common classical extraction techniques (CETs), including liquid-liquid extraction (LLE), solid-liquid extraction (SLE), Soxhlet extraction, and hydrodistillation, are very laborious, time-consuming, use high volumes of organic solvents, and in general, present low extraction efficiency and reproducibility. However, the success of FBCs analysis, identification, and characterization still strongly depends on the efficiency of the extraction method. Despite this reality, extraction technologies did not receive much attention until quite recently.

During the last two decades and to overcome the drawbacks of CETs, several new advanced extraction techniques, and more recently, emerging extraction techniques, emerged as highly efficient, faster, cheaper, and “greener” alternatives for the pre-treatment of complex samples (Figure 9).

Most of the prominent advanced extraction procedures include microextraction techniques (µETs), pressurized liquid extraction (PLE), PHWE, ultrasound-assisted extraction (UAE), and microwave-assisted extraction (MAE). In turn, emerging technologies includes supercritical (SFE) and subcritical fluid extraction (SbCE), electrotechnologies-based extraction such as pulsed electric fields (PEF) and high voltage electric discharge (HVED), and nanosorbent-based extraction techniques.

Among advanced extraction techniques, µETs experienced a wide range of applications and have become popular extraction approaches since they exhibit many advantages, such as simplicity, versatility, and high extraction efficiency, while being environmentally friendly. For these reasons, µETs have experienced promising developments and stimulated significant progress in laboratory sample treatment (Figure 10). The use of high throughput microextraction techniques, MEPS and µSPEed, for the determination of polyphenols was reported by Casado, et al. [126]. Pereira, et al. [127] reviewed the strengths and weaknesses of MEPS and the different MEPS architectures commercially available in the context of the MEPS applications. Additionally, innovative improvements were also highlighted, particularly those related to new applications and recent MEPS configurations and sorbents, such as the controlled directional flow or the innovative µSPEed variant [127]. The fundamentals, relevant improvements, and high-throughput applications of the QuEChERS method, which is also widely used with food matrices, was reviewed by Perestrelo, et al. [128]. Related with this, Casado, et al. [129] proposed an improved µQuEChERS protocol for the determination of 12 phenolic compounds in baby foods. Table 4 presents some applications of different extraction techniques used on polyphenols isolation.

In recent years, ultrasound-assisted extraction (UAE) has been one of the most employed advanced extraction techniques. UAE involves the use of US radiation in different devices such as sonoreactors, probes, water bath, etc., being the energy derived from sound waves propagated into the matrix. This facilitates the extraction of FBCs due to alterations produced in the cell wall of the matrix by bubble cavitation, with consequent improvement of the recovery of the target analytes, reduced extraction time, and decreased solvent consumption [143,144]. Low temperatures and extraction time are the main advantages relative to CETs. Zhou, et al. [145] proved the efficiency of the UAE technique for the extraction of natural antioxidants from fruits.

MAE is another efficient advanced approach tailored to FBCs extraction. It is based on the solvent heating caused by the interactions of the polar molecules in the media, which can be obtained by dipole rotation or ionic conduction [146]. This process facilitates the FBCs diffusion from the matrix to the solvent [146]. In MAE, extraction temperature and time, solvent composition, microwave power, and solid-liquid ratio are the most important parameters influencing the extraction efficiency. Regarding the solvents, water, methanol, acetone, ethanol, and different combinations of them are the most used in MAE procedure due to their higher power of dissipation by comparison to nonpolar solvents. Lower volumes of solvent and shorter extraction times constitute the major advantages of using MAE than CETs.

In terms of emerging extraction techniques, supercritical fluid extraction (SFE) holds great potential. In SFE, the solvent, most often CO_2_, is pumped under high pressures (above 1100 Psi) to a heating zone, where it is heated up to supercritical conditions. It is an environmentally friendly technique using a low amount of organic solvents and offering an increased selectivity towards some bioactive compounds. This procedure proved to be more effective then CETS to extract bioactive compounds from tea leaves [147].

Subcritical water extraction (SbFE) is a pressurized liquid extraction that uses water as a solvent, therefore increasing the mass transfer and extraction rate from the solid matrix. In SbFE, the sample in water solution is submitted to temperatures and pressures below the critical point of water. This offers a suitable, environmental, and cost-effective alternative to CETs, taking advantage of the water properties. Its use was investigated for the extraction of anthocyanins from fruits and vegetables [148] and a considerable amount of FBCs was obtained using SbFE in comparison with other extraction procedures not involving water or with the Soxhlet extraction. A similar extraction approach using subcritical carbon dioxide as a subcritical fluid was reported to extract FBCs from apple and peach pomaces [149]. However, this technique is limited to less polar compounds, not extracting some important flavonoids.

Regarding electrotechnologies, these are nonthermal techniques used in food technology for pasteurization and sterilization. They employ current that passes through samples to induce membranes electroporation and damage to the biological tissues. The electrical field creates a charge accumulation and a transmembrane potential able to cause electropermeabilization and the release of the target analytes. These techniques present several applications, namely in milk pasteurization [150], extraction of intracellular compounds [151], and fruit juice preservation [152].

The range of applications of advanced and emerging extraction techniques is very broad and includes environmental, agricultural, pharmaceutical, clinical, and forensics fields. In terms of target analytes, a vast number of them have been reported beyond FBCs and polyphenols, in particular (Figure 11).

The concentration, diversity, and bioactivity of the recovered FBCs are related to the extraction technique, while the choice of the technique is determined by the raw material, target analytes, cost of the procedure, and yield of the operation. As the extraction is not selective nor specific, additional steps are required to remove potential compounds including pigments, waxes, glycosides, fat, and sugars, among others, which can influence the instrumental analysis. For this purpose, SPE and dispersive SPE (d-SPE) are the commonly used clean up procedures. Primary secondary amine (PSA), typically used to remove pigments, sugars, organic acids, fatty acids and lipids; the octadecyl silica (C_18_), effective for the removal of highly lipid fractions; and graphitized carbon black (GCB), used to remove co-extracted pigments from pigmented matrices, such as green leafy vegetables, are the most popular agents to clean up the organic extracts [153]. Nevertheless, emerging sorbents are being developed and proposed. Hamed, et al. [154] proposed an innovative approach based on zirconium-based sorbent (Supel^TM^ QuE Z-Sep^+^) as a new material to remove fats and pigments more efficiently than PSA, C_18_, and GCB sorbents. Cabrera, Caldas, Prestes, Primel and Zanella [153] explored chitosan as an economic alternative to the C_18_ sorbent, keeping the efficiency of the clean up procedure in the extraction of veterinary drugs [155] and pesticides residues in milk [156]. To remove interfering substances from organic extracts of fish samples regarding the analysis of organochlorine pesticides, a sophisticated clean up procedure based on d-SPE (with PSA sorbent) followed by a dispersive liquid-liquid microextraction combined with the solidification of a floating organic droplet (DLLME-SFO) was proposed by Wang, et al. [157].

### Analytical Methods for Quantification

To obtain a deeper knowledge about the FBCs composition of different matrices, advanced and high-resolution analytical techniques are required. This includes high- and ultra-performance liquid chromatography (HPLC and UHPLC), which is the most universal technique for the separation, identification, and quantification of FBCs. Gas chromatography (GC) both coupled to mass spectrometry (MS), supercritical fluid chromatography (SFC), and capillary electrophoresis have also been successfully employed. These qualitative and quantitative analyses provide more specific and detailed information than spectrophotometric methods. FBCs analysis is commonly carried out in the reverse phase mode (RP-LC), using C-8 or C-18 based silica columns (10–30 mm L × 2 to 4 mm I.D.), although hydrophilic interaction liquid chromatography (HILIC) and bidimensional LC (LC × LC) have also been applied. To achieve high chromatographic resolution, several experimental parameters, such as gradient elution, mobile phase composition, stationary phase, and column temperature, must be optimized. The use of sub-2 µm particle packed columns combined with a high pressure system allows one to reduce analysis time as well as to increase resolution, efficiency, and sensitivity. The LC × LC system provides an increased selectivity and resolution in comparison with one dimensional LC for more complex matrices.

The elution is generally carried out using binary solvent systems, using methanol or acetonitrile (pure or acidified) as an organic solvent and acidified water (formic or acetic acids in percentages between 0.05–5% (*v*/*v*)) as a polar solvent. The acidification is necessary to minimize the peak tailing that suppresses the ionization of the phenolic hydroxyl groups.

The combination of LC with different types of detectors, including photodiode array detector (PDA), fluorescence detector (FD), and MS, allows one to obtain a comprehensive structural elucidation and identification of FBCs. Due to its robustness and low price, PDA is widely used, being useful for phenolic compounds identification (detected at 240–285 nm) in addition to flavones and flavonols (350–365 nm) and anthocyanins (460–560 nm). Although its highest selectivity and sensitivity achieving detection limits are much lower than the ones achieved using PDA, FD is less used in FBCs analysis as it requires the existence of fluorescence, like the one occurring in procyanidins and free catechins and epicatechins. Overall, the best performance is achieved using MS detectors employing an electrospray ionization either in positive or negative ion mode, although the best sensitivity is provided by negative mode. As analysers, simple quadrupole (Q), Triple quadrupole (QqQ), quadrupole time of flight (Q-TOF), and orbitrap-based hybrid mass spectrometry (LTQ-orbitrap) are the most popular. QqQ is broadly used for quantification purposes, being able to perform multiple reaction monitoring (MRM). Q-TOFMS presents the best resolution for identification of FBCs, allowing one to perform MS/MS experiments with improved structural information and selectivity. HPLC-ESI-QTOF/MS has been successfully used to identify phenolic compounds from different matrices, such as M. oleifera Lam [158], cranberry syrup [159], and avocado [160].

## 8. Remarks and Future Trends

Important challenges in FBCs applications will include deeper research regarding bioavailability and bioaccessibility to overcome the problems associated with their instability and low bioavailability. This will certainly boost more applications in the clinical environment. The molecular mechanisms and cellular signalling pathways in which different polyphenols act should be further investigated to potentiate their bioactivity. The potential side effects in addition to long-term toxicity constitute another interesting and important challenge to explore.

Due to the great diversity in the number and chemical nature of the bioactive compounds present in nature, their analysis and characterisation is a challenging task. To overcome some drawbacks regarding FBCs extraction and analysis/characterization, advanced and emerging extraction techniques, such as MAE, UAE, SFE, and SbFE, which are greener, faster, and more sensitive and reproducible than CETs, should be employed. However, the development of deeper and comprehensive high-resolution analytical techniques with improved sensitivity, efficiency, and selectivity is of utmost importance. UHPLC and LC × LC, namely when combined with MS, emerged as promising separation and characterisation techniques constituting high throughput platforms for highly selective, sensitive, fast, and high-resolution analysis and characterization of bioactive compounds.

Although the next-green extraction methods present a high extractive performance regarding polyphenols, the direct analysis capability of certain ambient MS methods have proven to be very useful for their direct analysis, identification, and characterization [161]. In this context, several relevant reports include DESI-MS for cosmetic additives [162]; LTP-MS, PSI-MS, and DART-MS of olive oil constituents [163,164]; and threadspray-MS for capsaicinoids [165]. In addition, Fatigante, et al. [166] coupled these techniques to form simplistic extraction processes for better performance, as well FCSI-MS on portable instruments for field analysis.

## Figures and Tables

**Figure 1 foods-10-00037-f001:**
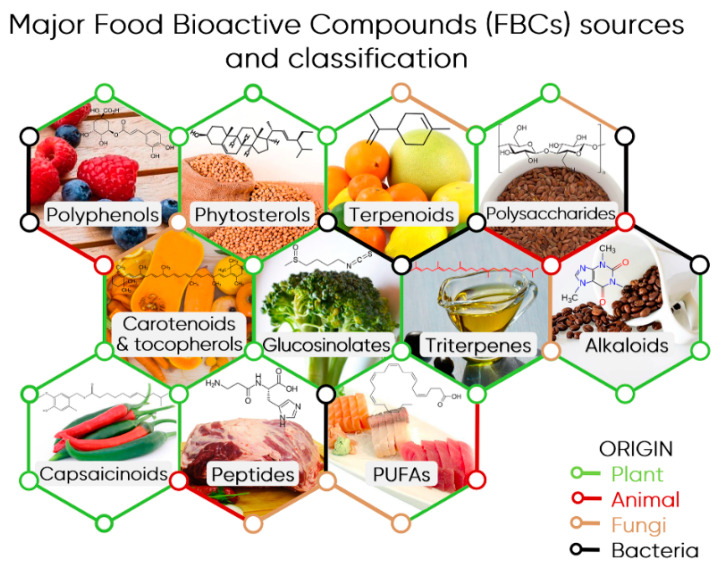
Major Food Bioactive Compounds (FBCs) sources and classification. An illustrative example of a source and compound is indicated for each class: polyphenols (chlorogenic acids in blueberry and raspberry fruits), phytosterols (stigmasterol in soybean), terpenoids (limonene in citrus fruits), polysaccharides (cellulose in flax seeds), carotenoids & tocopherols (β-carotene/vitamin A), glucosinolates (sulforaphane in broccoli), triterpenes (squalene from olive oil), alkaloids (caffeine in coffee beans), capsaicinoids (capsaicin in peppers), bioactive peptides (carnosine in red meat), and PUFAs (polyunsaturated fatty acids, docosahexaenoic acid—DHA, in different fishes).

**Figure 2 foods-10-00037-f002:**
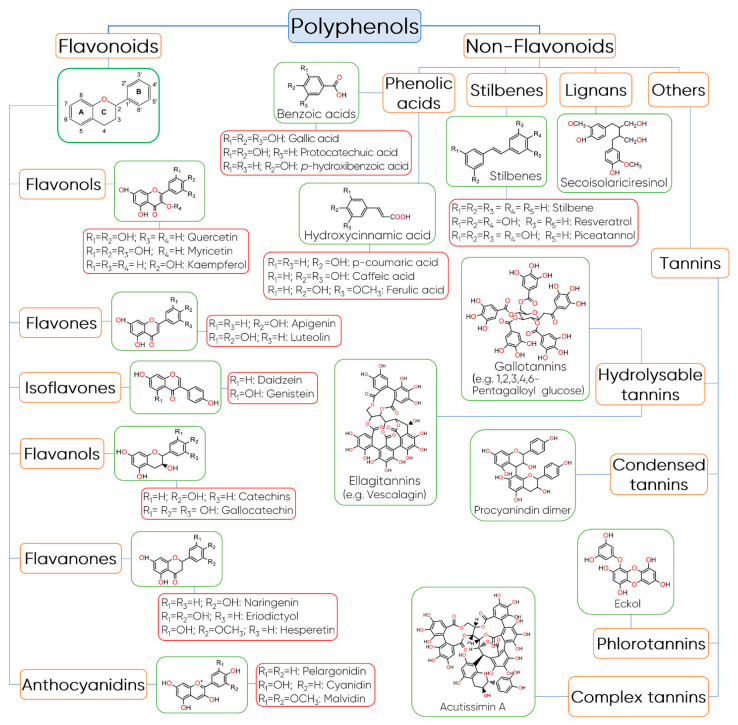
Classification and structure of polyphenols.

**Figure 3 foods-10-00037-f003:**
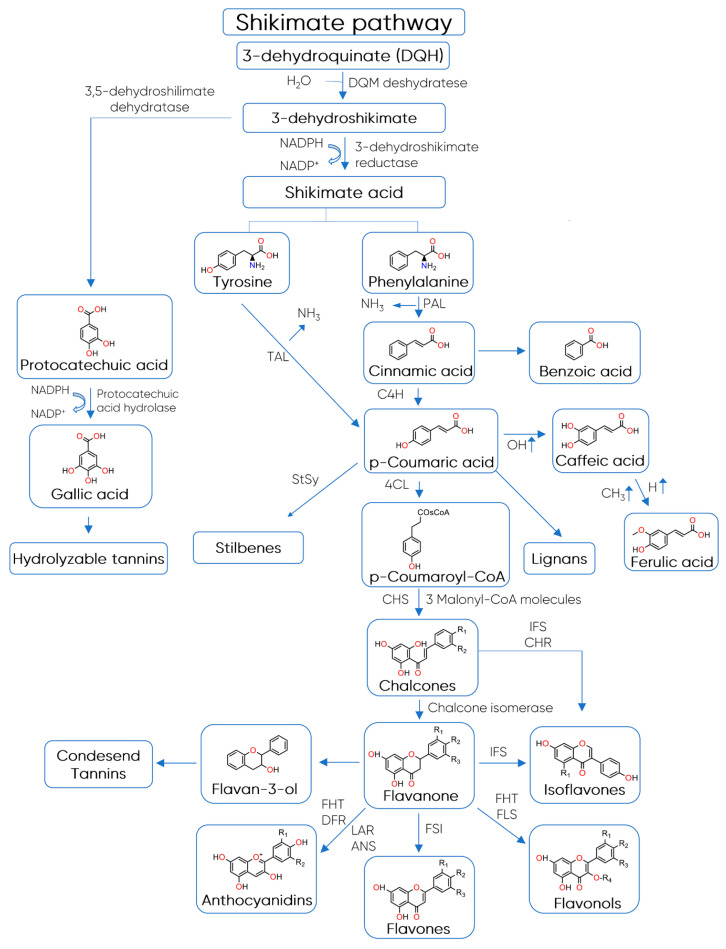
Biosynthesis of polyphenols. Abbreviations: 4CL—4-coumaryl:CoA ligase, ANS—anthocyanidin synthase, C4H—acid-4-hydroxylase, CHR—chalcone reductase, CHS—chalcone synthase, DFR—dihydroflavanone reductase, FHT—flavanone hydroxytransferase, FLS—flavonol synthase, FSI—flavanone synthase, IFS—isoflavanone synthase, LAR—leucoanthocyanidin synthase, PAL—phenylalanine ammonia lyase, StSy—stilbene synthase, TAL—tyrosine ammonia lyase (adapted from Fowler and Koffas [54] and de Araújo et al. [55]).

**Figure 4 foods-10-00037-f004:**
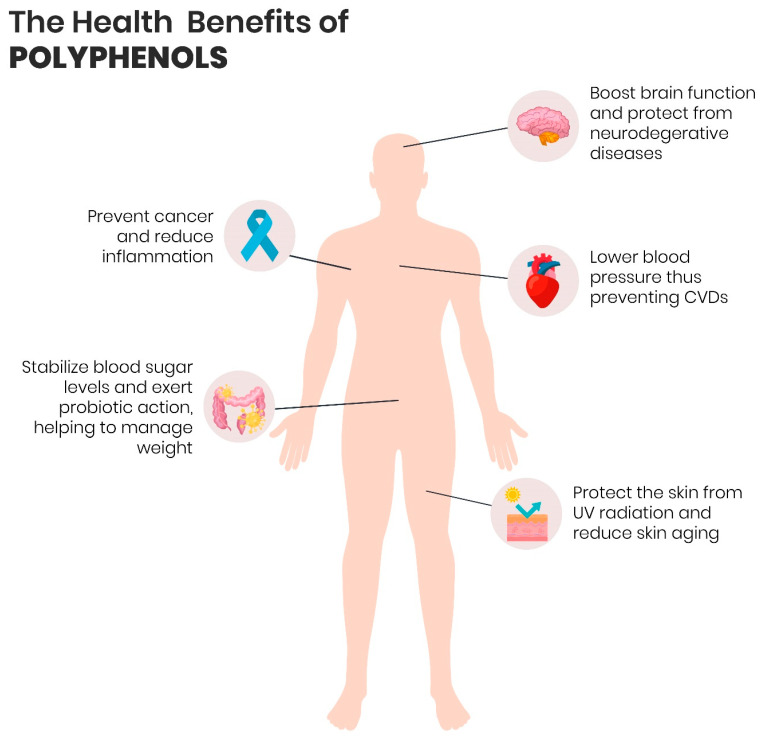
The potential benefits of dietary consumption of polyphenols for human health.

**Figure 5 foods-10-00037-f005:**
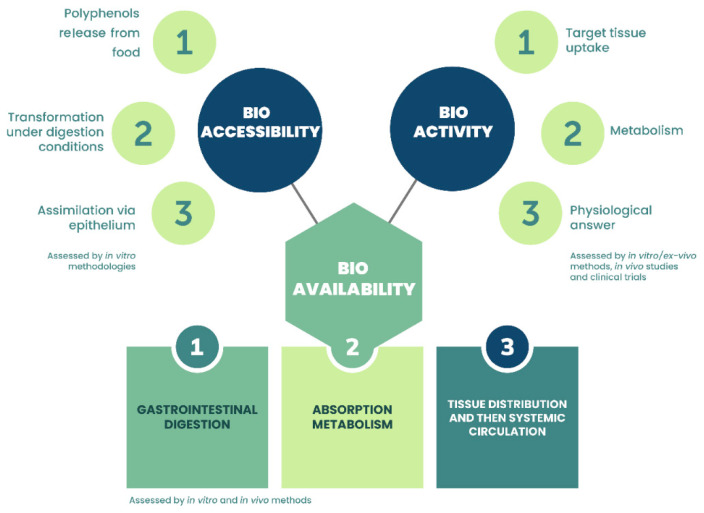
Events implicated in the bioavailability, bioaccessibility, and bioactivity of dietary polyphenols and the experimental approaches currently used for their assessment. Adapted from Thakur, et al. [102].

**Figure 6 foods-10-00037-f006:**
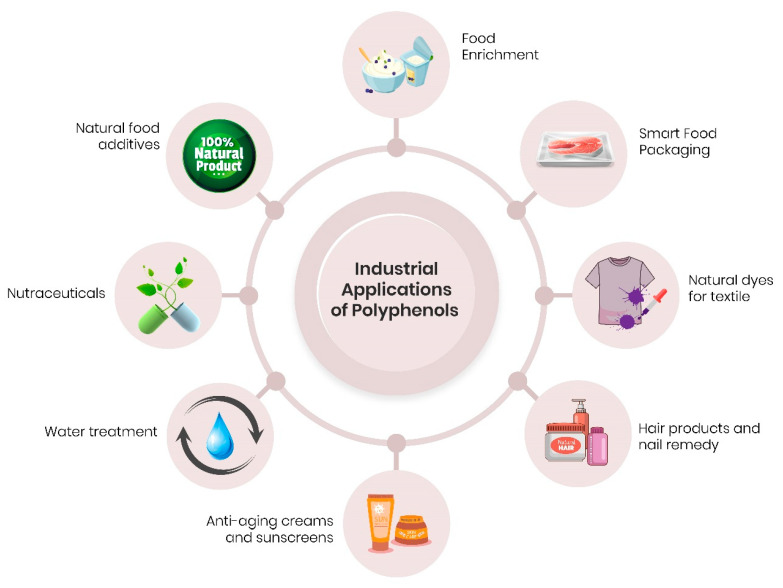
Traditional and emerging industrial applications of polyphenols in the development of products and/or technologies.

**Figure 7 foods-10-00037-f007:**
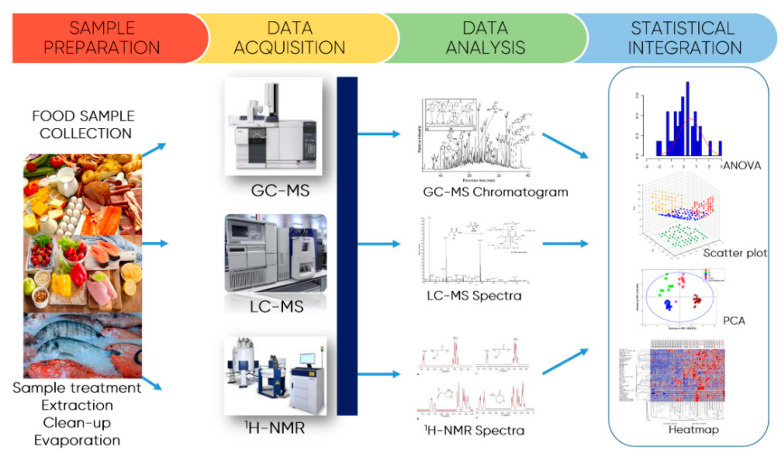
Different steps involved in the analysis of food and by-products from agro-food industry.

**Figure 8 foods-10-00037-f008:**
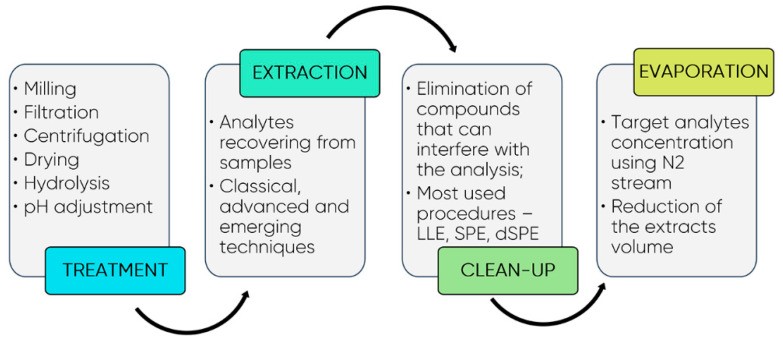
Strategies usually followed for the treatment and extraction of bioactive compounds from food matrices. Legend: dSPE—dispersive solid-phase extraction; LLE—liquid-liquid extraction; SPE—solid-phase extraction.

**Figure 9 foods-10-00037-f009:**
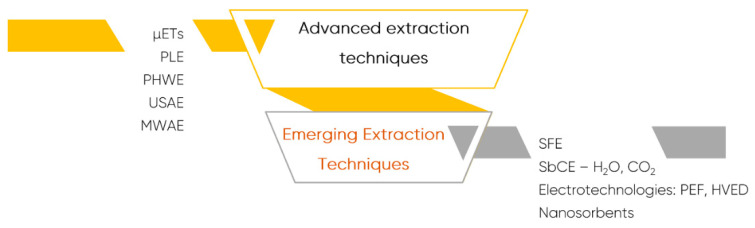
Advanced and emerging extraction techniques used in food analysis. Legend: HVED—high voltage electric discharge; MAE—microwave-assisted extraction; PEF—pulsed electric fields; PHWE—pressurized hot water extraction; PLE—pressurized liquid extraction; UAE—ultrasound-assisted extraction; SbCE—Subcritical extraction; SFE—supercritical fluid extraction; µETs—microextraction techniques.

**Figure 10 foods-10-00037-f010:**
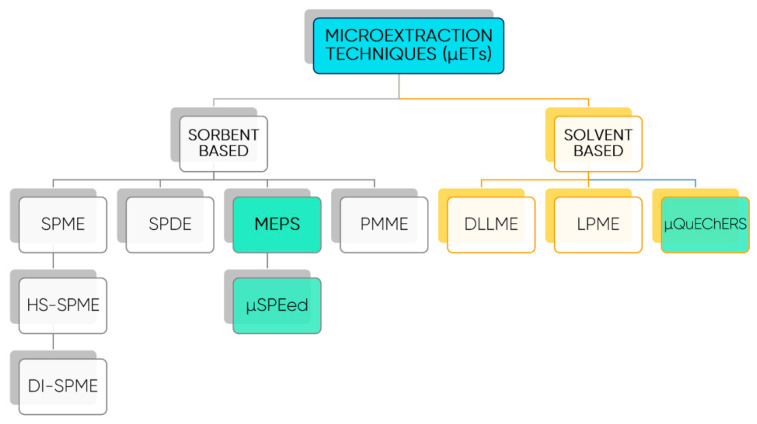
Sorbent and solvent-based microextraction techniques. Legend: SPME—Solid-phase microextraction; HS-SPME—Solid-phase microextraction in headspace mode; DI-SPME—Solid-phase microextraction in direct mode; MEPS—microextraction in packed syringe; µSPEed—miniaturization of MEPS; PMME—polymer monolith microextraction; DLLME—dispersive liquid-liquid microextraction; LPME—liquid phase microextraction.

**Figure 11 foods-10-00037-f011:**
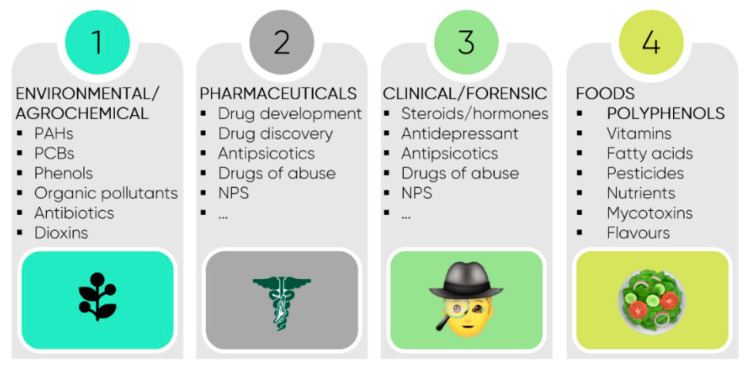
Potential application fields of advanced and emergent extraction techniques. Legend: NPS—Novel Psychoactive Substances; PAHs—polycyclic aromatic hydrocarbon; PCBs—Polychlorinated Biphenyls.

**Table 1 foods-10-00037-t001:** (**A**) Examples of polyphenols and phenolic extracts from plants with potential anticancer effects in vitro and in vivo. (**B**) Examples of the therapeutic approaches using polyphenols combined with anticancer drugs.

**(A)**							
**Polyphenols**		**Cancer Type**	**Type of Assay**				**Reference**
			**In Vitro Assay**		**In Vivo Assay**		
**Classes**	**Compounds**		**Cell-Lines**	**Concentration ***	**Model**	**Effects**	
Anthocyanins		Colorectal cancer	HT-29	400 μg/mL ^2^	Mouse model	↓ tumour size; ↓ inflammation	[72]
		Melanoma			Mouse model	Inhibited tumour growth and lung metastasis.	[64]
Flavanones	Naringin	Oesophageal cancer	YMI	354 μM ^1^	Mouse model	↓tumour size.	[68]
Flavones	Hispidulin	Acute myeloid leukaemia	MOLM-13	6.4 μM ^1^	-	-	[62]
			MV4-11	8.2 μM ^1^	-	-	[62]
	Luteolin	Lung carcinoma	A549	25–100 μM ^2^			[73]
		Acute myeloid leukaemia	MOLM-13	4.5 μM ^1^			[62]
			MV4-11	3.1 μM ^1^			[62]
	Tangeretin	Colorectal carcinoma	HCT116	12.5 μM ^2^	Mouse model	↓ tumour incidence; ↓ pathological symptoms	[74]
Flavonols	Kaempferol	Breast carcinoma			Mouse model	↓ Tumour size	[75]
Isoflavones	Alpinumisoflavone	Leukaemia	CEM/ADR5000	5.91 μM ^1^			[65]
		Breast carcinoma	MDA-MB-231-BCRP	65.65 μM ^1^			[65]
Lignans	Pycnanthulignene	Leukaemia	CEM/ADR5000	5.84 μM ^1^			[65]
		Colon carcinoma	HCT116	65.32 μM ^1^			[65]
**Phenolic acid**							
Stilbenes	Pterostilbene	Cervical cancer	TC1	15.61 μM ^1^	Mouse model	↓ Tumour size;	[76]
		Pancreatic cancer	-	-	Mouse model	Inhibited tumour growth	[77]
	Resveratrol	Cervical cancer	TC1	34.46 μM ^1^	Mouse model	↓ Tumour size;	[76]
		Melanoma	B16F10, A375 and B6	40 μM ^2^	-	-	[63]
		Lung cancer metastasis	-		Mouse model	↓ Angiogenesis; ↑ apoptosis metastatic colonies in the lungs	[63]
**Phenolic extracts from:**							
*Cynara cardunculus* L.		Oral carcinoma	SCC-25	184.81 μg/mL ^1^	-	-	[78]
Orange peel		Oesophageal cancer	YMI		Mouse model	↓ Tumour size.	
**(B)**							
**Compounds**		**Type of Cancer**	**Cancer Drug**	**Effect**			**Reference**
Quercetin		Lung cancer	Paclitaxel-resistant cells	Reversal of PTX resistance mitochondrial membrane potential MMP depolarization			[79]
Luteolin		Lung cancer	TRIAL	Positive effect synergetic			[67]
Orange peel extract		Oesophageal cancer	Doxorubicin	Decreased the side effects of chemotherapeutic treatment			[68]

Concentration *: ^1^—values expressed as the concentration required for 50% inhibition of cell proliferation (IC_50_ values), and ^2^—concentration that inhibits cell proliferation; TRIAL—tumour necrosis factor-related apoptosis-inducing ligand.

**Table 2 foods-10-00037-t002:** Effects of polyphenols on the cardiovascular system.

Polyphenols		Main Cardioprotective Effects	Type of Assay	Reference
Classes	Compounds			
Phenolic acids	Gallic acid	↑ Glucose tolerance	Animal model	[82]
	*p*-Coumaric acid	↑ Glucose tolerance	Animal model	[82]
Anthocyanins		↓ Platelet chemokines; ↓ LDL-C	Clinical trial	[70]
Flavan-3-ols	Catechins	↓ TC; ↓ LDL-C	Clinical trial	[71]
		↓ CK; ↓ CK-MB; ↓ LDH; ↓ cTnT; ↑ LVEF; ↓ LVIDs	Animal model	[83]
Flavanones	Hesperidin	↓ CK; ↓ cTnT; ↓ oxidative stress; ↓ cardiac tissue lesions	Animal model	[84]
Flavones	Apigenin	Anti-cardiac fibrosis	In vitro	[79]
Flavonols	Quercetin	Anti-platelet properties	In vitro	[85]
		↑ Vasodilatation	Clinical trial	[86]
	Kaempferol	Anti-platelet properties	In vitro	[85]
		↓ LDL-C	Clinical trial	[87]
Isoflavones		↓ Blood pressure	Clinical trial	[88]
Stilbenes	Resveratrol	↑ Antioxidant activity in the blood; ↓ Diabetic body weight	Clinical trial	[80]
		↓ TC; ↓ blood pressure; ↓ glucose;	Clinical trial	[89]
		↓ Blood glucose; ↓body weight; ↑ plasma insulin; ↓inflammation factors; ↓ oxidative stress; ↑ eNOS	Animal model	[81]
Curcuminoids	Curcumin	↓ Inflammation factors; ↓LDL-C; ↑ Nrf2, ↓ At1R; ↓ NF-kB	Clinical trial	[69]

Legend: At1R—Angiotensin II type 1 receptor; CK—creatine kinase; cTnT—cardiac troponin; eNOS—Endothelial nitric oxide synthase; LDH—lactate dehydrogenase; LDL-C low-density lipoprotein cholesterol; LVEF—ventricular ejection fraction; LVIDs—systolic internal diameter; NF-kB—Nuclear factor kappa B; Nrf2—Nuclear factor (erythroid-derived 2)-like 2; TC—Total cholesterol.

**Table 3 foods-10-00037-t003:** Some neuroprotective effects of polyphenols.

Polyphenols		Target Diseases	Neuroprotective Effects	Type of Assay	Reference
Classes	Compounds				
Phenolic acids	protocatechuic acid + chrysin (flavone)	PD	↓Neuroinflammation; ↓Dopamine neurons death; ↑NRF2 and NF-κB; ↓oxidative stress	In vitro/Animal model	[95]
Flavonols	Apigenin	AD	↓Neuroinflammation; ↓Apoptosis	In vitro	[96]
	Quercetin	Cognitive deficit associated with diabetes	↑Memory; ↓oxidative stress;	Animal model	[97]
Flavanone	Naringenin	PD	↓Oxidative stress; ↓Neuroinflammation ↑Motor performance	Animal model	[98]
Stilbene	Resveratrol	Chemobrain	Prevents cognitive decline	Animal model	[94]
Polyphenol extract	Main compounds				
White grape juice	Quercetin derivatives; proanthocyanidins;	Autoimmune encephalomyelitis; Multiple sclerosis	↓Neuroinflammation;	Animal model	[91]
Blueberry	Anthocyanins; phenolic acids; proanthocyanidins	PD	↓Dopamine neurons death	In vitro	[99]
Grape seeds	Proanthocyanidins	PD	↓Dopamine neurons death	In vitro	[99]
*Arabidopsis thaliana*	Phenolic acids, quercetin derivatives; kaempferol derivatives	AD	↓Neuroinflammation; ↓ oxidative stress; restored the locomotor activity	In vitro/animal model	[93]
Cherry juice	Anthocyanins	Mild-to-moderate dementia	↑Verbal fluency; ↑Shot- and long-term memory	Clinical trial	[92]

Legend: AD—Alzheimer’s disease; PD—Parkinson’s disease.

**Table 4 foods-10-00037-t004:** Selected examples of applications of emerging extraction techniques for the isolation of polyphenols from food samples.

Extraction Technique	Sorbent (Amount)	Food Matrix	Target Analytes	Methodology	Recoveries (%)	Ref.
**µ-QuEChERS**						
	MgSO_4_ (75 mg), PSA (12.5 mg)	Baby foods	12 polyphenols	UHPLC-PDA	71–100	[129]
**QuEChERS-USAE**						
	PSA (25 mg), C18 sorbent (25 mg), and MgSO_4_ (150 mg)	Fruits and Vegetables	12 polyphenols	UHPLC-PDA		[30]
**MEPS**						
	CMK-3 nanoporous carbon (2 mg)	Rosemary	Rosmarinic acid	HPLC-UV/VIS	94–106	[130]
	C18 (4 mg)	Beer	2 prenylflavonoids	UHPLC-PDA	67–100	[131]
	C8 (4 mg)	Wines	10 phenolic acids	UHPLC-PDA	77–100	[132]
**μSPEed**						
	PS/DVB-RP	Baby Foods	12 polyphenols	UHPLC-PDA	67–97	[133]
	PS/DVB-RP	Tea	8 polyphenols	UHPLC-PDA	89–103	[134]
**SPME**						
	PA fiber	Wines, Spirits, and Grape Juices	Resveratrol	HPLC-DAD	92–99	[135]
	VIED/MMF-SPME	Fruit juice	4 phenolic acids	HPLC–DAD	70–118	[136]
	PS-DVB-PAN	Grapes, Berries, and Wine	8 polyphenols	HPLC-TQ-MS/MS	69–82	[137]
**dSPE**						
	HMS-C18 (50 mg)	Fruit and vegetables juices and smoothies	20 Polyphenols	UHPLC-IT-MS/MS	57–99	[138]
**SBSE**						
	PDMS	Wine	6 stilbenes	GC-Q-MS	79–109	[139]
**μ-MSPD**						
	Florisil (150 mg)	Lime fruit	2 flavonoids	UHPLC-UV	90–96	[140]
	Middle-molecular-weight chitosan (25 mg)	Olive fruits	7 polyphenols	UHPLC-Q-TOF-MS/MS	80–113	[141]
	Silica-based C18 (200 mg)	Citrus fruit juice	7 flavonoids	HPLC-UV	86–94	[142]

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
