# Peer review of "Food Bioactive Compounds and Emerging Techniques for Their Extraction: Polyphenols as a Case Study"

_foods, 2020, doi:10.3390/foods10010037_

Round 1

Reviewer 1 Report

Food Bioactive compounds – importance and analytical methodologies  

This review present  a comprehensive and interesting information of the potential of polyphenols, from their chemical structure classification, biosynthesis to preventive effects on NCDs namely cancer, CVDs and NDDS, bioavailability and bioaccessibility and in addition show the most relevant industrial and environmental applications. The review show the advanced and emerging extraction techniques and high resolution analytical techniques used for polyphenol characterization, identification and quantification. The review is well organized and written and brings together important information in this field. However, it has some aspects that need to be reviewed before publication.

  1. The title is too general for the content of the review, as it focuses on phenolic compounds more than bioactive compounds and relation to analytical methodologies it focuses on emerging extraction strategies. The title needs to be revised and more in tune with the content included in the work.
  2. Section 7. Emerging extraction techniques and methodologies for polyphenols quantification.

This section is discussed in a general way and should include a table that summarizes current publications (last years) that include the use of the emerging extraction techniques described in this section (Figures 9 and 10) for polyphenol extractions.

  1. Review the text editing, as in Figures 9 and 10 the footer is cut into two pages.

Author Response

REVIEWER #1

This review present a comprehensive and interesting information of the potential of polyphenols, from their chemical structure classification, biosynthesis to preventive effects on NCDs namely cancer, CVDs and NDDS, bioavailability and bioaccessibility and in addition show the most relevant industrial and environmental applications. The review show the advanced and emerging extraction techniques and high resolution analytical techniques used for polyphenol characterization, identification and quantification. The review is well organized and written and brings together important information in this field. However, it has some aspects that need to be reviewed before publication.

The authors acknowledge the reviewer comments and suggestions

  1. The title is too general for the content of the review, as it focuses on phenolic compounds more than bioactive compounds and relation to analytical methodologies it focuses on emerging extraction strategies. The title needs to be revised and more in tune with the content included in the work.

Authors Ansewer:

According to the reviewer comment the title of the masnusript was revised in order to better express its content : Food Bioactive compounds and emerging techniques for their extraction: the polyphenols as case study

  1. Section 7. Emerging extraction techniques and methodologies for polyphenols quantification.

This section is discussed in a general way and should include a table that summarizes current publications (last years) that include the use of the emerging extraction techniques described in this section (Figures 9 and 10) for polyphenol extractions.

Authors Ansewer:

According to the reviewer comment a new table summarizing current publications on emerging extraction techniques for polyphenols was included on the manuscript

Review the text editing, as in Figures 9 and 10 the footer is cut into two pages.

Authors Ansewer:

According to the reviewer suggestion the figure captions were revised and corrected

Reviewer 2 Report

In this review article, the authors set out on the ambitious task to provide a snapshot of the efforts in the field of food-borne, bioactive chemical analysis, while also emphasizing the overall importance in the greater scientific discipline.    Overall, this is a very active research area, as seen by the timeliness of the references incorporated by the authors, as well as the diverse application areas that were of focus. 

I believe that the relevance of this research area coupled with the appreciable breadth of detail given (particularly in regards to current/emerging extraction techniques) will make this work a well-read and well-cited contribution to the literature.  This is a high quality review by renowned scholars with authority in the field, and I believe that this report will be of high interest to the readership of Foods. Only minor comments/recommendations remain prior to publication, and they are delineated below:

  • The authors provide an extensive and high-quality reference list, but it appears that a majority are included without the journal to which they were published. Enough citation information was provided to track down the references for review purposes, but this needs rectified prior to publication. 
  • Discussions involving the diverse application areas of polyphenols were very impactful, as well as the thorough discussion of current and emerging analytical techniques for polyphenol analysis. This will give the general scientific community a broad overview of this important area. While a majority of the stalwart analytical techniques were touched upon, I think an important area (perhaps falling within your Section 8: Remarks and Future Trends) is the newer “ambient” mass spectrometry (MS) techniques.  One aspect limiting the breadth of polyphenol screening/analysis in industrial, food safety/quality, forensics, and many other areas (as the authors dutifully report) is analytical throughput.  Next-gen extraction methods definitely help in this regard, but the direct analysis capability of certain ambient MS methods have proven useful in this area (and have been recently reviewed: see Brown et al., Anal. Methods., 2020, 12, 3974-3997). Reports pertinent to this application area include: DESI-MS for cosmetic additives (Nizzia et al., Anal. Methods, 2013, 5, 394-401), LTP-MS, PSI-MS, and DART-MS of olive oil constituents (Lara-Ortega et al., Talanta, 2018, 180, 168-175; Farre et al., Anal. Methods., 2019, 11, 472-482), threadspray-MS for capsaicinoids (Jackson et al., Anal. Chim. Acta., 2018, 1023, 81-88), to name a few.  There have been reports of coupling these techniques to simplistic extraction processes for better performance, as well, like FCSI-MS on portable instruments for field analysis (Fatigante et al., J. Am. Soc. Mass Spectrom., 2020, 31, 336-346). 

Author Response

In this review article, the authors set out on the ambitious task to provide a snapshot of the efforts in the field of food-borne, bioactive chemical analysis, while also emphasizing the overall importance in the greater scientific discipline.Overall, this is a very active research area, as seen by the timeliness of the references incorporated by the authors, as well as the diverse application areas that were of focus. 

I believe that the relevance of this research area coupled with the appreciable breadth of detail given (particularly in regards to current/emerging extraction techniques) will make this work a well-read and well-cited contribution to the literature.  This is a high quality review by renowned scholars with authority in the field, and I believe that this report will be of high interest to the readership of Foods. Only minor comments/recommendations remain prior to publication, and they are delineated below:

The authors acknowledge the reviewer comments and suggestions

  • The authors provide an extensive and high-quality reference list, but it appears that a majority are included without the journal to which they were published. Enough citation information was provided to track down the references for review purposes, but this needs rectified prior to publication.

Authors Ansewer:

According to the reviewer suggestion the references wre reviewed and corrected

  • Discussions involving the diverse application areas of polyphenols were very impactful, as well as the thorough discussion of current and emerging analytical techniques for polyphenol analysis. This will give the general scientific community a broad overview of this important area. While a majority of the stalwart analytical techniques were touched upon, I think an important area (perhaps falling within your Section 8: Remarks and Future Trends) is the newer “ambient” mass spectrometry (MS) techniques. One aspect limiting the breadth of polyphenol screening/analysis in industrial, food safety/quality, forensics, and many other areas (as the authors dutifully report) is analytical throughput. Next-gen extraction methods definitely help in this regard, but the direct analysis capability of certain ambient MS methods have proven useful in this area (and have been recently reviewed: see Brown et al.,  Methods., 2020, 12, 3974-3997). Reports pertinent to this application area include: DESI-MS for cosmetic additives (Nizzia et al., Anal. Methods, 2013, 5, 394-401), LTP-MS, PSI-MS, and DART-MS of olive oil constituents (Lara-Ortega et al., Talanta, 2018, 180, 168-175; Farre et al., Anal. Methods., 2019, 11, 472-482), threadspray-MS for capsaicinoids (Jackson et al., Anal. Chim. Acta., 2018, 1023, 81-88), to name a few.  There have been reports of coupling these techniques to simplistic extraction processes for better performance, as well, like FCSI-MS on portable instruments for field analysis (Fatigante et al., J. Am. Soc. Mass Spectrom., 2020, 31, 336-346)

Authors Ansewer:

According to the reviewer suggestion a sentence was added to the manuscript highlighting the importance of MS techniques:

Although the next-green extraction methods present a high extractive performance towards polyphenols,  but the direct analysis capability of certain ambient MS methods have proven very useful for its direct analysis, identification and characterization [146]. Several pertinent reports in this context includes, DESI-MS for cosmetic additives [147] LTP-MS, PSI-MS, and DART-MS of olive oil constituents [148,149], and threadspray-MS for capsaicinoids [150]. In addition, Fatigante et al. [151], reports on coupling these techniques to simplistic extraction processes for better performance, as well FCSI-MS on portable instruments for field analysis [151].”